# Wind/Wave Testing of a 1:70-Scale Performance-Matched Model of the IEA Wind 15 MW Reference Wind Turbine with Real-Time ROSCO Control and Floating Feedback

Matthew Fowler [1,*], Eben Lenfest [1], Anthony Viselli [1], Andrew Goupee [2], Richard Kimball [2], Roger Bergua [3], Lu Wang [3], Daniel Zalkind [3], Alan Wright [3] and Amy Robertson [3]

1   Advanced Structures and Composites Center, University of Maine, Orono, ME 04469, USA; eben.lenfest@composites.maine.edu (E.L.); anthony.viselli@composites.maine.edu (A.V.)
2   Department of Mechanical Engineering, University of Maine, Orono, ME 04469, USA; agoupe91@maine.edu (A.G.); richard.w.kimball@maine.edu (R.K.)
3   National Wind Technology Center, National Renewable Energy Laboratory, Arvada, CO 80007, USA; roger.bergua@nrel.gov (R.B.); lu.wang@nrel.gov (L.W.); daniel.zalkind@nrel.gov (D.Z.); alan.wright@nrel.gov (A.W.); amy.robertson@nrel.gov (A.R.)
*   Correspondence: matthew.fowler@composites.maine.edu; Tel.: +1-207-581-2236

**Abstract:** Experimental results from the Floating Offshore-wind and Controls Advanced Laboratory (FOCAL) experimental program, which tested a performance-matched model of the IEA Wind 15 MW Reference Turbine on a 1:70 scale floating semisubmersible platform, are compared with OpenFAST simulations. Four experimental campaigns were performed, and data from the fourth campaign, which focused on wind and wave testing of the scaled floating wind turbine system, are considered. Simulations of wave-only, wind-only, and wind/wave environments are performed in OpenFAST, and results for key metrics are compared with the experiment. Performance of the real-time Reference OpenSource COntroller (ROSCO) in above-rated wind conditions, including the effects of the floating feedback loop, are investigated. Results show good agreement in mean values for key metrics, and hydrodynamic effects are matched well. Differences in the surge resonant behavior of the platform are identified and discussed. The effect of the controller and floating feedback loop is evident in both the experiment and OpenFAST, showing significant reduction in platform pitch response and tower base bending load near the platform pitch natural frequency.

**Keywords:** floating wind; offshore wind; wind turbine; turbine controls; model testing; floating feedback; numerical simulation; OpenFAST; ROSCO

## 1. Introduction

Pursuit of more economical floating offshore wind turbines continues to drive innovation in platform designs and turbine control systems. Advances in analysis capabilities create opportunities to optimize designs and improve the design process. One such methodology is controls co-design (CCD), by which the impact of controls is considered and leveraged throughout the design stages to optimize the system. The U.S. Department of Energy's Advanced Research Projects Agency–Energy (ARPA-E) has funded the ATLANTIS research program focused on bringing control co-design to floating offshore wind. The program supported projects in three areas: analysis code development, experimental data, and novel technologies. One such project was the Floating Offshore-wind and Controls Advanced Laboratory (FOCAL) experimental program, which aimed to create the first public model-test dataset that considered a 1:70 scale 15 MW floating offshore wind turbine that included advanced turbine controls, floating hull load mitigation technology, and hull flexibility. The FOCAL program consisted of a series of four model-scale floating offshore wind turbine experimental campaigns in the University of Maine's Alfond Wind-Wave

Ocean Engineering Laboratory (W2). Two cross-collaborative efforts during the project supported validation of various software tools from the ATLANTIS program using data from the first three FOCAL experiments, focusing on the aerodynamic performance of the wind turbine in a fixed-bottom configuration as well as hydrodynamic performance of the floating system with tuned-mass dampers using a flexible tower and fixed mass topside. Data from the four campaigns have been uploaded upon completion of the experimental work and are publicly available with the research community [1].

Providing open datasets on floating turbine systems is a challenging proposition. Often, turbine and/or platform designs are proprietary, and the cost of performing a basin- scale model test can be significant. Similar to the DeepCwind [2], INNWIND.EU [3], and COREWIND [4] programs, the FOCAL project specifically chose open designs for the turbine and platform with the goal of providing an accessible and applicable dataset for floating offshore wind. There are several methodologies for generating datasets to validate designs or modeling tools, and they can generally be characterized by how the wind and wave loads are realized [5]. When considering the turbine, in order of increasing fidelity, the loading can be represented by a fixed mass, a drag disk, a simple actuator system to represent global loading, or a fully operational wind turbine or actuated system that strives to represent the fully coupled interaction between the turbine, controller, and platform. Hybrid testing methods, where a component of the system is represented through a hardware-in-the-loop implementation using a characterized numerical model, are gaining popularity. This approach has been used to simulate the response of the turbine through actuators, typically with actuated tendons [6] or fans [7,8], with promising results. An alternate hybrid approach can instead include the hydrodynamic effects of the floating system through an actuated system in conjunction with a wind turbine scale model in a wind field, see [9,10]. A summary of scale-model testing campaigns through 2014 can be found in [11], and the current state of the art for hybrid testing methods is discussed within the COREWIND project [12], which considered multiple ways to model the IEA Wind 15 MW turbine in wind/wave basin tests. The InnWind program [13] performed testing of a 10 MW floating wind turbine on a semisubmersible using a performance-matched turbine and a ducted fan to simulate the thrust force. Bachynski et al. [14] present hybrid model tests of the NREL 5 MW [15] turbine on a semisubmersible platform at 1:30 Froude scale, and Thys et al. [16] discuss testing of the DTU 10 MW reference turbine on a semisubmersible at 1:36 scale. In these campaigns, the turbine was represented by the MARINTEK Real-Time Hybrid Model (ReaTHM™) method [6], which used a series of cables connected to the rotor-nacelle assembly to affect 5 degrees of freedom forcing in rotor thrust, tangential force, generator torque, and pitch and yaw bending moments. The system could effectively actuate loads at frequencies up to 2 Hz, covering most frequencies of interest for typical floating wind turbine tests. Fontanella et al. [8] present testing of the NREL 5 MW turbine on the DeepCwind semisubmersible at 1:50 scale using hardware-in-the-loop and an array of fans to simulate the turbine. Industry standard control strategies using a proportional-integral controller collective blade pitch controller in the above-rated conditions, as well as an individual blade pitch control algorithm, were considered. Cao et al. [17] conducted 1:64-scale tests of the DTU 10 MW reference turbine on a semi-submersible using a performance-matched turbine. Generator torque and collective blade pitch controllers were considered, but in above-rated conditions, the pitch angle of each blade was set manually, and an external control loop maintained constant rotor speed. Bredmose et al. [18] tested the Triple Spar floater with a performance-matched model of the DTU 10 MW reference turbine at 1:60 scale. Three control strategies were considered, including fixed blade pitch, a standard land-based controller, and a tuned controller designed to address floating system instability. The platform pitch instability inherent in floating turbine control was demonstrated with the land-based controller, and reductions were realized through retuning the controller. Most recently, the IEA Wind 15 MW turbine was tested at 1:70 scale on a version of the VolturnUS-S platform at the Coastal Ocean and Sediment Transport laboratory using real-time hybrid testing to simulate

the turbine [19]. Machine learning was used to train a surrogate model using OpenFAST simulations to control a ducted fan in the hardware-in-the-loop system to generate the turbine thrust load. The simulations included a number of simplifications, most notably fixed rotor speed and blade pitch, which implies that effects of turbine controls are not considered.

The FOCAL program utilized a performance-matched wind turbine styled after the IEA Wind 15 MW reference turbine [20], including the Reference OpenSource Controller (ROSCO) [21] turbine controller, in conjunction with a Froude-scale floating platform and corresponding environments. This was performed to investigate the aerodynamic load effects of controller actions in the coupled floating system and evaluate performance of the ROSCO controller in-the-loop with the scale model turbine where it had authority over the blade pitch and generator torque set points. To date, fully coupled wind/wave experimental results using a performance-matched turbine with controls are not available for the IEA Wind 15 MW wind turbine.

This paper considers the performance of the floating wind turbine system and presents a comparison of OpenFAST [22] simulation results against experimental data for above-rated conditions, including the effects of the floating feedback control loop in ROSCO. FAST [23], and subsequently OpenFAST, has a long history of open-source development and has been validated with numerous experimental campaigns [24–26]. With the recent development of structural control elements to represent the tuned-mass damper (TMD) elements, as well as ROSCO, OpenFAST was used extensively in the FOCAL program to design the scale rotor, implement and tune ROSCO, and conduct model validation efforts of aerodynamic performance from Campaign 1, hydrodynamic performance from Campaigns 2 and 3, and the fully floating model from Campaign 4. This paper is organized with the OpenFAST methodology and experimental setup in Section 2. Section 3 presents results from the simulation and comparisons with the experiment. Significant findings are discussed in Section 4, and conclusions are presented in Section 5.

## 2. Materials and Methods

The goal of the FOCAL project was to generate validation data for analysis codes; the project team has iteratively developed an OpenFAST model over the course of the FOCAL project. This model is included in the dataset and shared with the research community as a starting point for representing the FOCAL system. Data from Campaign 1 defined the aerodynamic characteristics of the as-built rotor, as discussed in [27]. Results from Campaign 2/3 were used to define the hull mode and tuned-mass damper representation, as described in [28]. The subject of this paper is the Campaign 4 model, which combines the turbine topside and the floating platform into a fully coupled model. It was created by combining the Campaign 1 and Campaign 2/3 OpenFAST models, with modifications to represent as-built component mass/inertia values and configuration changes, such as adjusting tower flexibility and replacing the dummy mass with an operational turbine.

### 2.1. Model Description

Control of the turbine in both the experiment and the OpenFAST model was achieved through ROSCO. As explained in [29], ROSCO was implemented in real time for the experiment where data from the laboratory were scaled from model scale to full scale and passed to the full-scale ROSCO using the appropriately scaled time step. This means that, effectively, the same ROSCO control algorithm was used for both the experiment and the full-scale simulation, with the exception of the time step. The basin loop time was effectively 0.001 s model scale, which corresponded to 0.008 s full scale, while the integration time step for the OpenFAST analysis was 0.025 s.

As described in [30], the experiment used a performance-matched turbine that was designed using the Selig-Donovan SD7032 series airfoil. Smoothed lift and drag polars from the LIFES50+ project [31] were used in the baseline design, as shown in Figure 1. In

the figure, "Unmodified" refers to the original lift and drag data, and "Baseline" refers to the smoothed version used as input into the blade design tool.

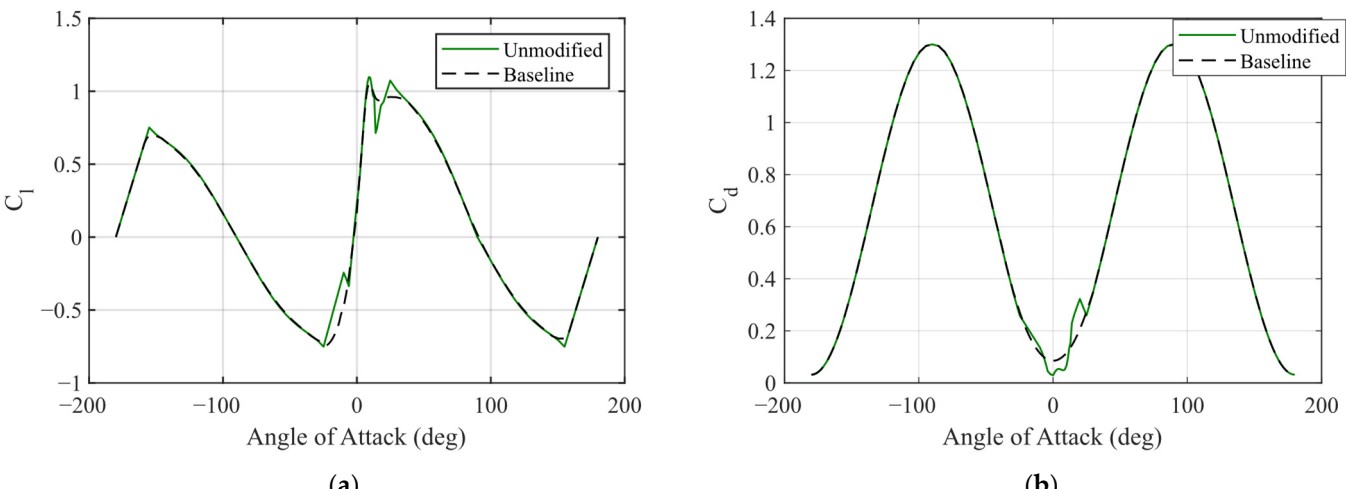

**(a)**　　　　　　　　　　　　　　　　**(b)**

**Figure 1.** (**a**) SD7032 lift coefficient vs. angle of attack; (**b**) SD7032 drag coefficient vs. angle of attack.

In Campaign 1, the as-built airfoil performance was determined from a series of open-loop control tests where the wind speed was held constant. The rotor speed and blade pitch were independently incremented, the turbine was allowed to reach a steady state, and the resulting rotor thrust and torque were measured to obtain the rotor performance over a range of tip speed ratios and angles of attack. These data were summarized as Cp, Ct, and Cq surfaces. This process was performed for two different wind speeds, one near the rated wind speed and one at a higher wind speed, to account for Reynolds effects, which were determined to be significant over the range of operating wind speeds in the experiment. The lift and drag polars in the OpenFAST model were then tuned to match the measured rotor performance using an optimization routine that altered the lift and drag coefficients within specific parameter bounds with the objective of matching the measured rotor performance. The baseline, rated tuned, and above-rated tuned polars are shown in Figure 2, where "Baseline" refers to the smoothed lift and drag polars used as input to the rotor design tool, and "Final" designates the tuned polars from the optimization process. The tuned polars were also included in the FOCAL dataset. The Cp, Ct, and Cq surfaces were also used to retune ROSCO, creating a new set of gains for the controller run both in the experiment and in OpenFAST. This process was repeated for both sets of performance data, representing the two wind speeds. Additionally, the floating feedback gain was tuned using the ROSCO toolbox and the as-built floating system properties. This process is described in more detail in publications from the FOCAL project [1].

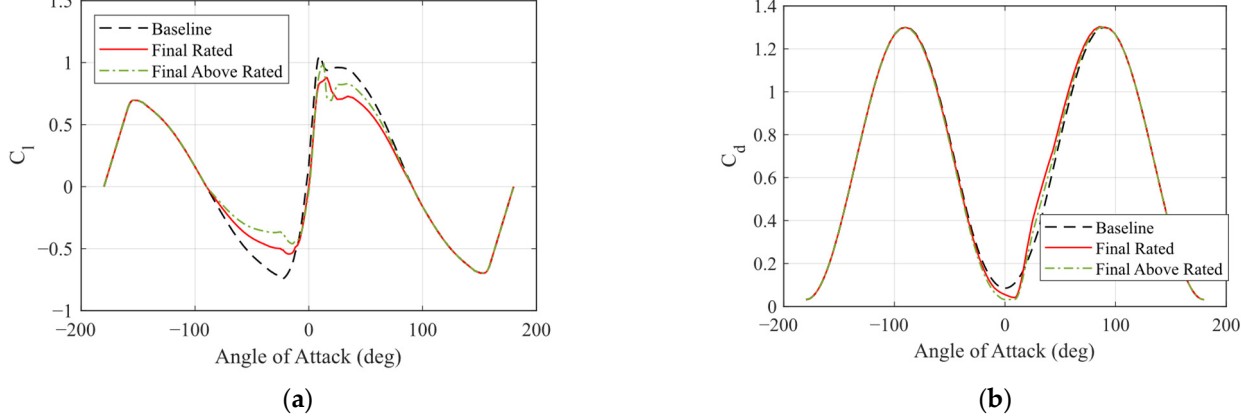

**(a)**　　　　　　　　　　　　　　　　**(b)**

**Figure 2.** (**a**) Tuned lift coefficient vs. angle of attack; (**b**) Tuned drag coefficient vs. angle of attack.

The main parameters of the model are published with the data set and summarized in Table 1. Note that all data are presented at full scale, following Froude-scaling relationships.

**Table 1.** Main parameters of the floating turbine model.

| Property | Unit | Value |
|---|---|---|
| Total System Properties ($XYZ_1$ Coordinate System *) | | |
| Mass | kg | $2.073 \times 10^7$ |
| Center of gravity (CG)–Height from keel | m | 19.1 |
| CG–X offset | m | 0.2 |
| CG–Y offset | m | −0.5 |
| Ixx (Roll Inertia) about system CG | kg·m$^2$ | $4.937 \times 10^{10}$ |
| Iyy (Pitch Inertia) about system CG | kg·m$^2$ | $4.972 \times 10^{10}$ |
| Hull Properties ($XYZ_2$ Coordinate System *) | | |
| Mass | kg | $1.866 \times 10^7$ |
| CG–Height from Keel | m | 6.9 |
| CG–X Offset | m | 0.1 |
| CG–Y Offset | m | −0.6 |
| Ixx (Roll Inertia) about hull CG | kg·m$^2$ | $1.353 \times 10^{10}$ |
| Iyy (Pitch Inertia) about hull CG | kg·m$^2$ | $1.402 \times 10^{10}$ |
| Izz (Yaw Inertia) about hull CG | kg·m$^2$ | $1.52 \times 10^{10}$ |
| Rotor Nacelle Assembly Properties ($XYZ_3$ Coordinate System *) | | |
| Mass | kg | $1.197 \times 10^6$ |
| CG–Height from keel | m | 168.7 |
| CG–X Offset | m | −4.4 |
| Ixx (Roll Inertia) about RNA CG | kg·m$^2$ | $6.751 \times 10^8$ |
| Iyy (Pitch Inertia) about RNA CG | kg·m$^2$ | $3.723 \times 10^8$ |

* Coordinate systems are defined in Figure 3.

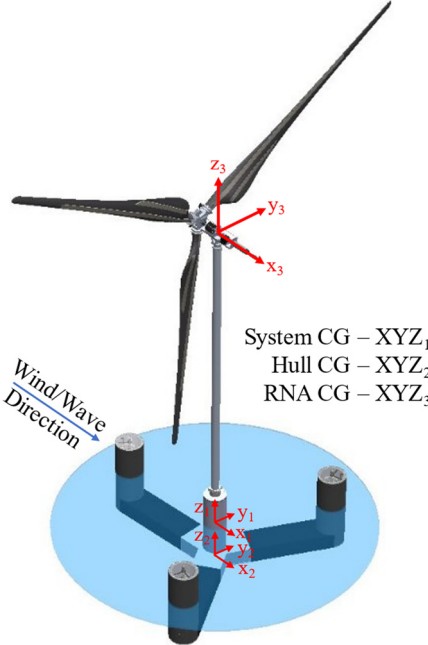

**Figure 3.** Coordinate systems for system properties.

The tuned-mass damper elements are represented in the OpenFAST model but are not considered in this comparison. They are modeled with stiff springs in the OpenFAST model, and comparisons are made to experiment conditions where the TMDs were similarly fixed in place.

## 2.2. Wind Environment

This study considers one of the three wind environments that were run in the experimental campaign: a turbulent wind environment (designated W03) at an above-rated mean wind speed of 24.05 ms$^{-1}$. Wind calibration dwell measurements of this environment were taken without the model in the basin and using one anemometer that was located at the hub center, one placed at hub height but offset laterally by 70 m, which is roughly 60% of the rotor radius, and one that was offset laterally and downwind. Additionally, a survey of the rotor plane was carried out and used to determine the averaged mean wind speed and uniformity of the rotor. To represent both the correct wind turbulence and rotor averaged mean velocity, the wind time history for the OpenFAST model is created by scaling the dwell time history to match the mean of the rotor averaged survey by multiplying the time series by the appropriate factor. This time history is then implemented as a spatially uniform unsteady wind in the OpenFAST model. Wind calibration time series are shown in Figure 4, while the power spectral density (PSD) plot uses data from 2000 to 9500 s for consistency with the analysis in this paper.

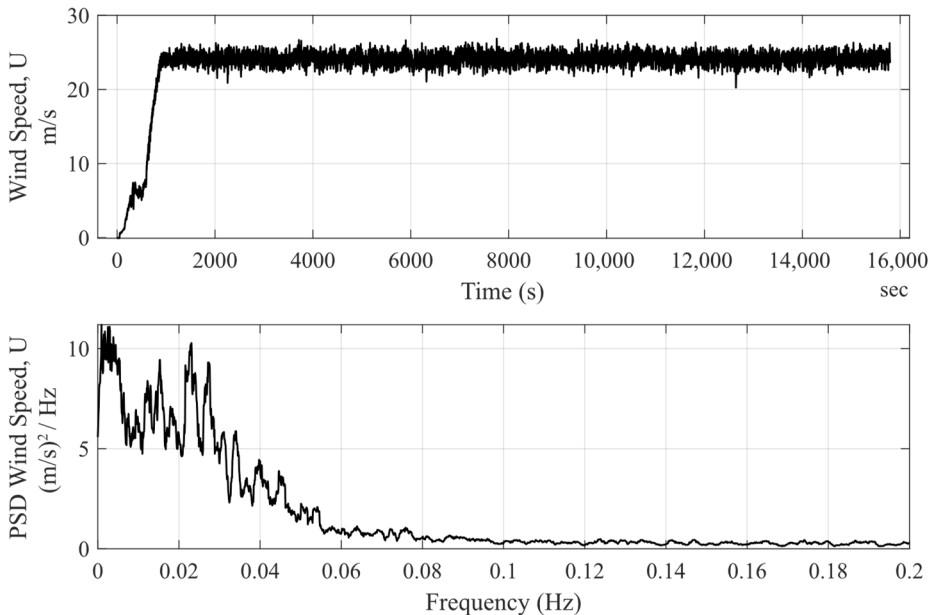

**Figure 4.** Time history and PSD of U component of wind velocity.

## 2.3. Wave Environment

In the experiment, a JONSWAP spectrum irregular wave definition was considered for a 1-year extreme sea state corresponding to rated wind conditions. Five realizations of this environment were created that had different time histories but the same statistics, identified as E21–E25. Five repeats of the E21 wave environment were run in the experiment to assess repeatability, and this wave environment is what is considered in the OpenFAST model. Specifications for this environment are given in Table 2.

**Table 2.** Wave environment parameters.

| Wave ID | Significant Wave Height, $H_s$ [m] | Peak Period, $T_P$ [s] | $\gamma$ [-] |
|---------|-----------------------------------|------------------------|--------------|
| E21–E25 | 8.1 | 12.8 | 2.75 |

In the experiment, waves were calibrated by running each wave in the wave tank without the model installed. Wave time history data were collected from an array of probes that were located where the model would be installed. Data from these runs were used to create the time history wave environment for the OpenFAST model. Figure 5 shows

the wave elevation time series and PSD plot where the PSD is calculated using data from 2000 to 9500 s for consistency.

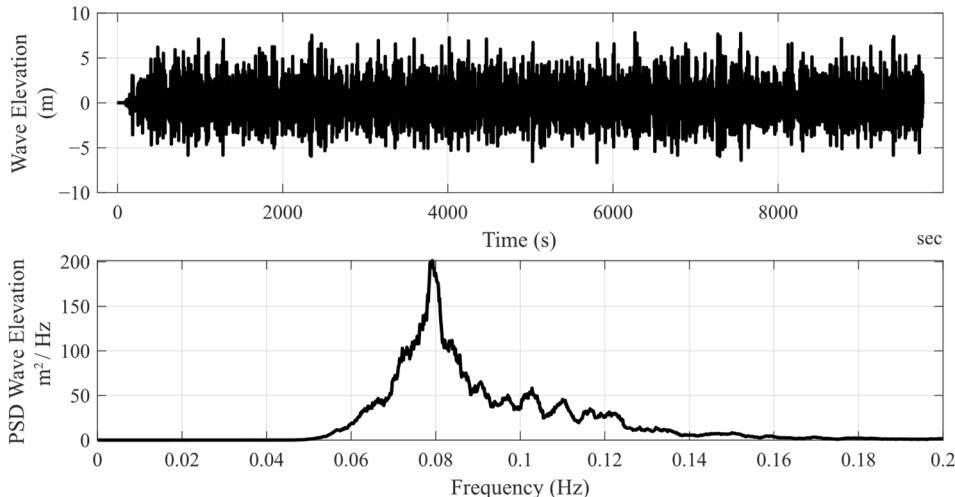

**Figure 5.** Time history and PSD of wave elevation.

*2.4. Mooring*

The mooring system for the experiment consisted of three lines, each extending radially from the outer columns and spaced 120° apart. These lines were nominally horizontal and mounted to anchors near the same elevation as the fairlead. Each line was stiff monofilament in line with a linear spring. The stiffness of each line was determined such that the surge restoring force of the mooring system was equivalent to the VolturnUS-S catenary chain mooring system in the mean displaced position due to rated thrust load on the turbine. There was also a restoring effect due to the umbilical cable, which acted as an additional soft mooring line. This effect was quantified by performing offset tests in the surge and sway direction, both with and without the umbilical cable attached, and results with the umbilical cable attached are shown in Table 3.

**Table 3.** Linearized mooring stiffnesses.

| Degree of Freedom | Experiment | OpenFAST |
|---|---|---|
| Surge [N·m$^{-1}$] | 217,863 | 211,420 |
| Sway [N·m$^{-1}$] | 224,008 | 221,232 |

The mooring system was modeled in OpenFAST using three lines with properties shown in Table 4. Note, Line3 is the lead line while Line 1 and Line 2 are the port and starboard lines, respectively. The three lines had the same nominal stiffness, but the bow line was shorter in the experiment. The umbilical cable's effect is included as additional linear surge and heave stiffness terms, as shown in Section 3.1.

**Table 4.** Mooring model parameters.

| Line ID | Line Location | EA [N] | Unstretched Length [m] | Number of Segments [-] |
|---|---|---|---|---|
| 1 | Port | 33,862,500 | 268.75 | 15 |
| 2 | Starboard | 33,816,290 | 266.27 | 15 |
| 3 | Bow (Lead Line) | 12,285,240 | 99.88 | 15 |

*2.5. Instrumentation*

The turbine model was instrumented to measure turbine performance, global dynamics, mooring loads, and internal loading, as shown in Figure 6. Table 5 compares OpenFAST

results with data from the experiment. In the experiment and the simulation, ROSCO was run at full scale, and all comparisons are completed using full-scale values (including time).

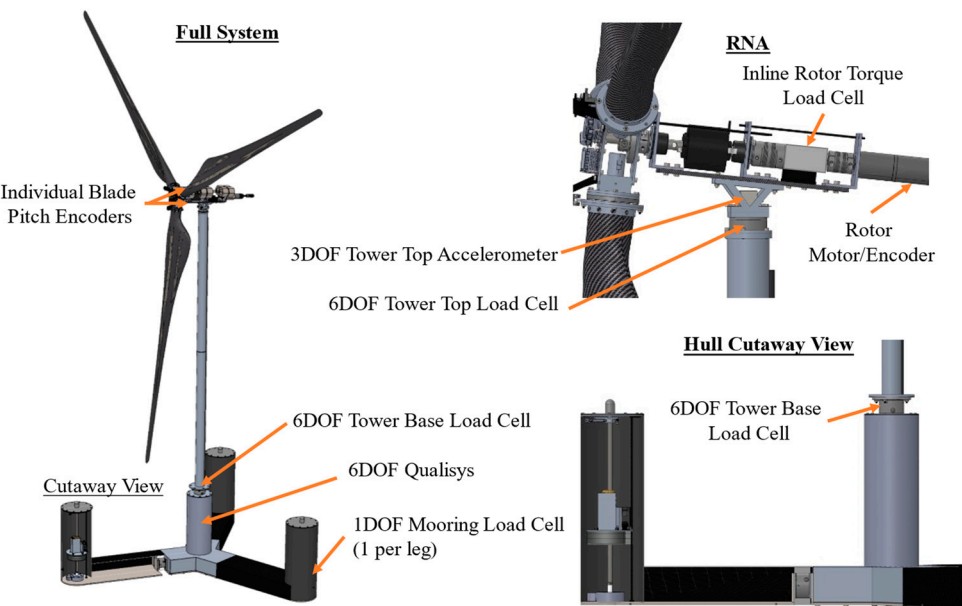

**Figure 6.** Instrumentation layout (DOF = degree of freedom).

**Table 5.** Instrumentation and data channel list.

| Channel | Experiment Source | OpenFAST Channel | Notes |
|---|---|---|---|
| Platform Surge | Qualisys 6DOF Motion | PtfmSurge | Rigid body motion |
| Platform Pitch | Qualisys 6DOF Motion | PtfmPitch | Rigid body motion |
| Rotor Torque | Inline Torque Sensor | RotTorq | Low-speed shaft torque |
| Rotor Speed | Rotor Encoder | RotSpeed | 1:1 Gearbox |
| Blade Pitch | Blade Pitch Encoder | BldPitch1 | Collective blade pitch |
| Tower Base Moment | 6DOF Load Cell | TwrBsMyt | Pitch moment |
| Mooring Fairlead Tension | 1DOF Inline Load Cell | FAIRTEN3 | Lead line tension |

## 3. Results

### 3.1. Free Decay

Free-decay tests of the model were performed in its floating configuration with moorings attached, both with and without the umbilical cable. This was performed to quantify the effect of the umbilical cable on the model. For this comparison, only the cases with the umbilical cable attached are considered. The additional linear stiffness terms are then computed for surge and heave to match the free-decay period and were small compared with the mooring restoring and hydrostatic stiffness terms, respectively. The additional linear and quadratic damping values computed are shown in Table 6 and are consistent with the p-q method outlined in [32].

**Table 6.** Additional linear stiffness, linear damping, and quadratic damping terms.

| Degree of Freedom | | Linear Stiffness | Linear Damping | Quadratic Damping |
|---|---|---|---|---|
| Surge | [N·m$^{-1}$ or N·(m·s$^{-1}$)$^{-1}$ or N·(m·s$^{-1}$)$^{-2}$] | $6.9 \times 10^3$ | $1.6 \times 10^5$ | $1.0 \times 10^6$ |
| Sway | [N·(m·s$^{-1}$)$^{-1}$ or N·(m·s$^{-1}$)$^{-2}$] | N/A | $1.6 \times 10^5$ | $1.0 \times 10^6$ |
| Heave | [N·m$^{-1}$ or N·(m·s$^{-1}$)$^{-1}$ or N·(m·s$^{-1}$)$^{-2}$] | $3.2 \times 10^4$ | 0.0 | $6.6 \times 10^6$ |
| Roll | [N·m·rad$^{-1}$ or N·m·(rad·s$^{-1}$)$^{-2}$] | N/A | $2.7 \times 10^8$ | $7.2 \times 10^{10}$ |
| Pitch | [N·m·rad$^{-1}$ or N·m·(rad·s$^{-1}$)$^{-2}$] | N/A | $2.7 \times 10^8$ | $7.2 \times 10^{10}$ |
| Yaw | [N·m·rad$^{-1}$ or N·m·(rad·s$^{-1}$)$^{-2}$] | N/A | 0.0 | $7.2 \times 10^{10}$ |

The resulting natural periods are shown in Table 7.

**Table 7.** Natural periods for experiment and simulation.

| Degree of Freedom | Experiment | | OpenFAST | |
|---|---|---|---|---|
| | Period * [s] | Frequency [Hz] | Period * [s] | Frequency [Hz] |
| Surge | 80.8 | 0.0124 | 79.2 | 0.0126 |
| Sway | 79.6 | 0.0126 | 78.1 | 0.0128 |
| Heave | 21.2 | 0.0473 | 20.6 | 0.0486 |
| Roll | 31.1 | 0.0322 | 30.1 | 0.0333 |
| Pitch | 30.8 | 0.0325 | 30.0 | 0.0333 |
| Yaw | 50.94 | 0.0196 | 46.05 | 0.0217 |

\* Periods are from free-decay tests performed in still water with no wind and do not include the aerodynamic effects of an operating turbine in wind.

### 3.2. Above Rated Conditions

The test environment considered is a turbulent wind at an above-rated wind speed and an irregular sea state, referred to as W03 and E21, respectively. In the experiment, wave condition E21 was performed five times to determine repeatability and estimate measurement uncertainty. This comparison uses average statistics from these five runs to compare with the OpenFAST simulation. In the experiment, the tests performed included wind-only, wave-only, and then combined wind/wave loading. These cases are also simulated in OpenFAST to compare the wind-only, wave-only, and coupled loading in the model. Results are generally compared in context with different frequency regions, namely those dominated by wind turbulence, system resonance responses, and wave energy. These regions are summarized in Table 8 and follow those used by Wang et al. [29]. Results from higher frequencies are not shown in spectral plots to focus on where ROSCO is most active but are still considered in calculating statistics. To remove startup transients and compare results, power spectral results and statistics are presented for the time subset of 2000 to 9500 s with the mean value removed. The equilibrium values of the system in still water with no wind applied are also removed prior to computing statistics; therefore, the results can be interpreted as the deviation from the still water equilibrium condition.

**Table 8.** Key frequency ranges for PSD analysis.

| Frequency Range [Hz] | Dominant Forcing or Response |
|---|---|
| 0.0005–0.0120 | Wind Turbulence |
| 0.0120–0.0500 | Platform Pitch Response |
| 0.0550–0.2500 | Wave Energy |

### 3.3. Wave Only

The wave-only condition is simulated using the wave elevation time history data collected from the calibration of the wave environments during the experiment (OpenFAST option WaveMod = 5). The simulation is run for 9754 s and analysis is performed from 2000 to 9500 s to remove startup transients and provide consistent comparison. During these cases, the turbine is parked with the blade pitch at $0°$. Second-order wave kinematics are not computed, but sum- and difference-frequency loading is computed from the second-order WAMIT [33] quadratic transfer functions (0.12 s and 0.12d files, respectively). There is no current or inclusion of Morison-type elements, and the additional damping is included through the linear and quadratic damping matrices, as discussed in Section 3.1.

Key metrics are considered in the frequency domain, shown as PSD results in Figure 7, where "Exp, R#" refers to experimental results from the five repeated runs (R1 to R5), and "OF" refers to OpenFAST results. Statistics are presented in Table 9, where results for the experiment are the average of each statistic over all repeated runs, and OpenFAST results are statistics for the single OpenFAST simulation. The wave-only results show that the

hydrodynamic forcing on the model is well accounted for. Because the turbine is parked, comparison of turbine operation is not discussed here. Mean, range, and standard deviation (SD) of platform surge and pitch motion, as well as tower base pitch bending moment and lead fairlead tension compare favorably. The experiment exhibits more surge motion at the surge natural frequency of 0.0124 Hz while platform pitch motion is well represented, both in the wave forcing region as well as at the platform pitch natural frequency of 0.0325 Hz. Discussion of this is provided in Section 4.1. Fairlead tension follows the same trend as the surge, with a good match on mean values and larger variation at the surge natural frequency in the experiment.

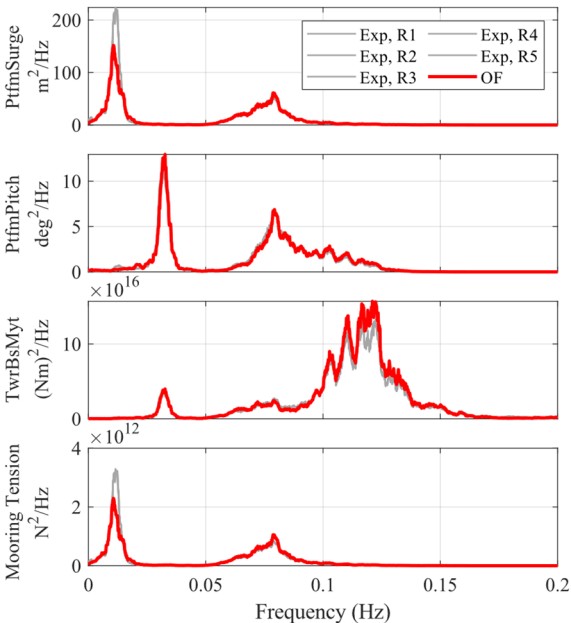

**Figure 7.** Wave-only PSD results.

**Table 9.** Wave-only statistics.

| Channel | Unit | Experiment, Average | | | OpenFAST | | |
|---|---|---|---|---|---|---|---|
| | | Mean | Range | SD | Mean | Range | SD |
| Pltfm Surge | m | 0.6 | 11.7 | 1.45 | 0.6 | 11.6 | 1.39 |
| Pltfm Pitch | deg | 0.0 | 3.6 | 0.46 | 0.0 | 3.3 | 0.46 |
| TowerBsMyt | N·m $\times 10^8$ | −0.01 | 5.46 | 0.66 | −0.01 | 5.22 | 0.70 |
| Lead Fairlead | N $\times 10^6$ | 0.07 | 1.48 | 0.18 | 0.07 | 1.54 | 0.18 |

*3.4. Wind Only*

The wind-only condition is simulated using time history wind data from the calibration of the wind environment for the experiment. The wind is represented as a spatially uniform and unsteady wind field, with uniform directionality and no swirl. The rotor is operational during the wind conditions and under the baseline ROSCO control. The wind speed is slowly ramped up from 0 ms$^{-1}$ to the above-rated condition, as shown in Figure 4, and ROSCO is utilized to control the rotor torque and blade pitch of the turbine to regulate operation. The simulation is run for 15,794 s and analysis is conducted from 2000 to 9500 s to remove startup transients and provide consistent comparison.

Results from the wind-only condition show good agreement of mean values for key metrics, as shown in Table 10. For global motion, there is a larger mean platform surge and pitch in the experiment. Figure 8 shows that the dynamic motion near the surge natural frequency is well matched; however, platform pitch response near the platform pitch natural frequency is larger for the OpenFAST model.

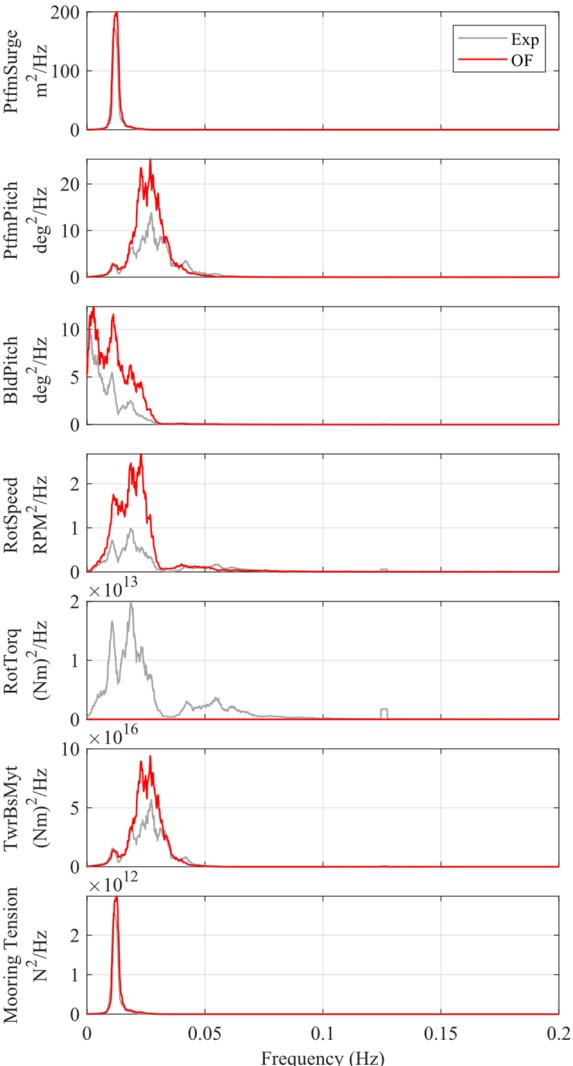

**Figure 8.** Wind-only PSD results.

**Table 10.** Wind-only statistics.

| Channel | Unit | Experiment, Average | | | OpenFAST | | |
|---|---|---|---|---|---|---|---|
| | | Mean | Range | SD | Mean | Range | SD |
| Pltfm Surge | m | 8.8 | 4.9 | 0.76 | 7.0 | 4.9 | 0.82 |
| Pltfm Pitch | deg | 5.2 | 2.7 | 0.44 | 4.3 | 4.2 | 0.55 |
| Blade Pitch | deg | 17.1 | 1.9 | 0.31 | 17.0 | 3.2 | 0.43 |
| Rotor Speed | RPM | 7.46 | 1.1 | 0.13 | 7.56 | 1.5 | 0.20 |
| Rotor Torque | N·m $\times 10^6$ | 18.67 | 5.93 | 0.68 | 18.67 | 0.00 | 0.00 |
| TowerBsMyt | N·m $\times 10^8$ | 3.72 | 2.86 | 0.42 | 3.12 | 2.82 | 0.37 |
| Lead Fairlead | N $\times 10^6$ | 0.78 | 0.61 | 0.10 | 0.78 | 0.61 | 0.10 |

Mean blade pitch is well matched while there is a larger range and standard deviation in the OpenFAST model and more energy in the blade pitch actuation, rotor speed, platform pitch, and tower base bending moment in the frequency range between 0.0 and 0.05 Hz, corresponding to wind turbulence and platform pitch. While mean rotor torque agrees between the experiment and OpenFAST, there is no rotor torque variation in the OpenFAST simulation, as ROSCO utilizes constant generator torque in the above-rated region. In contrast, the experiment utilized a proportional-integral controller on the scale turbine motor torque to track the constant rotor torque set point from ROSCO. The implications of

this are further explored in Section 4.2. Lastly, the mooring response agrees well in both mean value and dynamic response.

*3.5. Combined Wind and Wave*

3.5.1. Baseline ROSCO

Combined wind and wave results were simulated for the E21 wave condition with the corresponding W03 wind condition. The simulation was run for 9754 s based on the length of the wave time history. The wind was ramped up from 0 ms$^{-1}$ with the turbine operational under the baseline ROSCO's control, referred to as "RO." Analysis was performed from 2000 to 9500 s to remove startup transients. Results of the OpenFAST simulation are compared with the average of the five experimental cases that were performed.

Results from the combined cases show similar trends as the wind-only and wave-only cases. With regard to mean values, the experiment exhibits larger platform surge and pitch motion, and OpenFAST continues to underpredict surge motion at the natural frequency, as shown in Table 11.

**Table 11.** Wind/wave statistics with baseline ROSCO (RO).

| Channel | Unit | Experiment, Average | | | OpenFAST | | |
|---|---|---|---|---|---|---|---|
| | | Mean | Range | SD | Mean | Range | SD |
| Pltfm Surge | m | 9.7 | 12.0 | 1.61 | 7.5 | 9.2 | 1.33 |
| Pltfm Pitch | deg | 5.2 | 4.6 | 0.59 | 4.3 | 4.7 | 0.67 |
| Blade Pitch | deg | 17.1 | 2.0 | 0.32 | 17.0 | 3.1 | 0.44 |
| Rotor Speed | RPM | 7.46 | 1.3 | 0.15 | 7.56 | 1.7 | 0.21 |
| Rotor Torque | N·m $\times 10^6$ | 18.67 | 6.05 | 0.72 | 18.67 | 0.00 | 0.00 |
| TowerBsMyt | N·m $\times 10^8$ | 3.74 | 6.04 | 0.79 | 3.11 | 5.54 | 0.74 |
| Lead Fairlead | N $\times 10^6$ | 1.07 | 1.51 | 0.20 | 0.84 | 1.24 | 0.17 |

Differences in rotor performance exhibit the same trends as in the wind-only case. The mean blade pitch is matched, while the OpenFAST simulation has more blade pitch and rotor speed variation below 0.05 Hz, and more platform pitch motion and tower base bending around the platform pitch frequency, as shown in Figure 9. Results in the wave energy range from 0.055 Hz to 0.25 Hz generally agree well, with the experiment showing slightly more platform pitch motion and tower base bending around the wave peak frequency of 0.078 Hz. The double-peaked response in tower base loading for semisubmersibles is well documented [34] and the difference seen here at the wave peak frequency is likely due to the additional pitch motion seen in the experiment at these frequencies. The dominant contributions to the tower base bending moment at the higher frequencies are well represented by both. Mooring tensions follow the trend of surge motion, where the experiment has larger mean value and more energy at the surge resonant frequency.

3.5.2. ROSCO with Floating Feedback

This wind/wave case was also run using the floating feedback control option in ROSCO, designated as "FL" in Figure 10. For this case, floating feedback was enabled for platform pitch rotational velocity (OpenFAST DISCON option Fl_Mode = 2). In the experiment, the platform pitch rotational velocity was provided to ROSCO at the Froude-scaled time step of the basin real-time controller, whereas in OpenFAST, it is computed internally. The remaining ROSCO parameters were identical between the experiment and OpenFAST ROSCO implementation. The simulation was run for 9754 s with analysis performed for 2000 to 9500 s. To quantify the effect of the floating feedback control, results are presented comparing the experimental results with and without feedback as well as comparing OpenFAST with and without feedback.

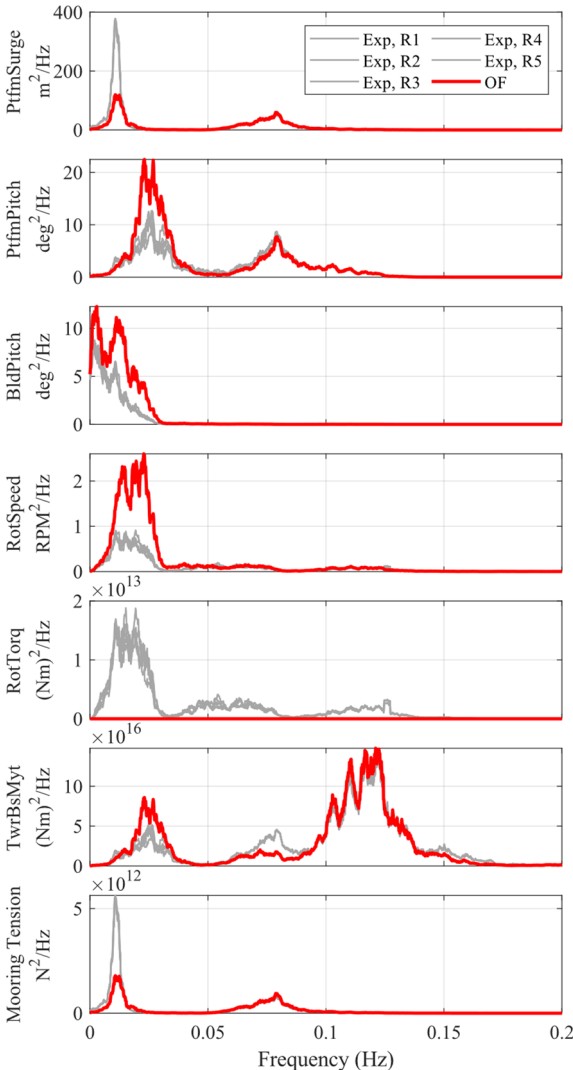

**Figure 9.** Wind/wave with baseline ROSCO (RO) PSD results.

Table 12 shows percent differences calculated such that positive values indicate an increase in the metric for the FL case and negative values are a decrease in the metric. The results show that mean values for key metrics are largely unaffected by the floating feedback control loop in both the experiment and OpenFAST. Surge and mooring dynamic responses are similarly unaffected; however, noticeable changes in the dynamic performance for other metrics are observed. Platform pitch and tower base moment range and SD decreased in both the experiment and OpenFAST, while overall blade pitch actuation increased. Rotor torque variation also increased for the experiment while OpenFAST maintained a constant rotor torque, as identified in Section 3.4.

Examining these parameters in the frequency domain, Table 13 compares integrals of various metrics over frequency ranges corresponding to wind energy, platform pitch motion, and wave frequency, and values are computed such that positive values indicate an increase in the metric for the floating feedback control. The effect of the floating feedback can be clearly seen in the platform pitch frequency range of 0.012–0.05 Hz, where there is a decrease in platform pitch, rotor speed, and tower base moment, as shown in Table 13 and Figure 10. In this range, there was a 59% reduction in platform pitch energy and a 61% decrease in tower base moment in the OpenFAST model. In the experiment, there were 32% and 38% decreases in pitch motion and tower base bending, respectively. Further investigating the blade pitch response in Figure 10, there is a clear reduction in blade pitch actuation between 0.012 and 0.023 Hz for both the experiment and OpenFAST, whereas,

above this frequency, there is additional energy that only exists when floating feedback is enabled. The increased blade pitch actuation at these higher frequencies is in the wave energy range and is an undesired effect due to limitation in filtering the floating feedback signal, as discussed in Section 4.3. This effect is shown in both the experiment as well as the OpenFAST model.

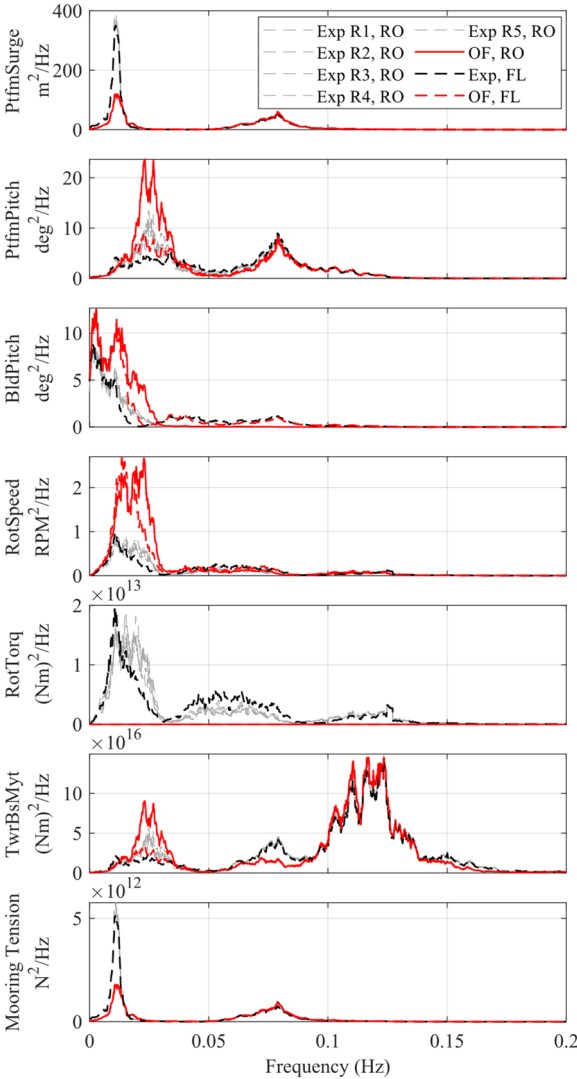

**Figure 10.** Wind/wave with floating feedback (FL) PSD results.

**Table 12.** Wind/wave statistics comparison, effect of floating feedback.

| | Experiment, % Difference | | | OpenFAST, % Difference | | |
|---|---|---|---|---|---|---|
| **Channel** | **Mean** | **Range** | **SD** | **Mean** | **Range** | **SD** |
| Pltfm Surge | 3% | 1% | −2% | 0% | 1% | 0% |
| Pltfm Pitch | 0% | −12% | −4% | 0% | −13% | −15% |
| Blade Pitch | 0% | 22% | 13% | 0% | 6% | 0% |
| Rotor Speed | 0% | −1% | 2% | 0% | −6% | −6% |
| Rotor Torque | 0% | 7% | 4% | 0% | 0% | 0% |
| TowerBsMyt | 0% | −8% | −2% | 0% | −2% | −5% |
| Lead Fairlead | 3% | 1% | −2% | 0% | −1% | 0% |

**Table 13.** Wind/wave PSD integral comparison, effect of floating feedback.

| Channel | Experiment | | | OpenFAST | | |
|---|---|---|---|---|---|---|
| | Wind Freq * | Pitch Freq * | Wave Freq * | Wind Freq * | Pitch Freq * | Wave Freq * |
| Pltfm Surge | −5% | −1% | −2% | 3% | 0% | −1% |
| Pltfm Pitch | 20% | −32% | 7% | 12% | −59% | 9% |
| Blade Pitch | −1% | 6% | 182% | 2% | −35% | 183% |
| Rotor Speed | 18% | −26% | 26% | 16% | −25% | 17% |
| Rotor Torque | 28% | −23% | 25% | N/A | N/A | N/A |
| TowerBsMyt | 23% | −38% | −3% | 14% | −61% | −1% |
| Lead Fairlead | −6% | −2% | −2% | 3% | −1% | 0% |

\* Frequency ranges for PSD integrals shown in Table 8.

## 4. Discussion

### 4.1. Surge Resonant Response

In the wave-only condition, the experiment exhibits more surge motion at the surge natural frequency of 0.0124 Hz. The presented OpenFAST model does not include drag elements and, therefore, does not consider excitation due to viscous effects. This underprediction of surge resonant response is consistent with prior work, including work validating FAST with basin data, e.g., [25,35]. Gueydon et al. [36] found the effects of second-order difference frequency wave forcing to be significant on the surge resonant response of the OC4 semisubmersible. Subsequent work to capture this response more accurately includes high-fidelity computational fluid dynamics (CFD) as well as specific improvements to mid-fidelity codes, such as OpenFAST, to better represent the viscous loading on the floating platform. Kvittem et al. [37] showed that quadratic drag coefficients tuned from free-decay tests underpredicted the surge resonant response, while better agreement was obtained when calibrating drag coefficients from irregular wave test data. Berthelsen et al. [38] tuned drag coefficients from free-decay tests, and then further tuned the values in the splash zone to better match irregular wave test data. Pegalajar and Bredmose [39] presented an operational modal analysis methodology for tuning drag coefficient from irregular sea state test data and observed that the coefficients generally increased with sea state severity. Wang et al. [32] discussed tuning drag coefficients from CFD analysis and also demonstrated an improved response in OpenFAST through tuning a depth-dependent drag coefficient and implementing vertical wave stretching in [40]. It is suggested that including viscous drag elements and tuning the drag coefficients, as well as including wave stretching, be carried out to better represent the viscous effects on the floating platform.

Moving to the wind-only condition, the response at resonance agrees well, whereas the mean surge response was larger in the experiment. The turbulent wind environment has energy in the low-frequency range that could directly excite surge resonant motion; therefore, differences in the realized wind time history could affect the response. In the wind/wave condition, the surge resonant response in the experiment was greater than that of the wave-only and wind-only responses. In contrast, in the OpenFAST model, the response in the wind/wave condition exhibits less motion than either the wave-only or wind-only conditions. As Coulling et al. [41] discuss, the surge resonant response is highly dependent on damping, so it is possible that the low-frequency aerodynamic damping forces are not well matched between the experiment and OpenFAST when the turbine is moving due to the wave environment. Further work to investigate the magnitude and phasing of the aerodynamic loads relative to the viscous loading on the platform is suggested to further explore this phenomenon.

*4.2. Effect of ROSCO above Rated Torque Control*

Differences in the blade pitch response can be partially explained by investigating the operation of the ROSCO torque controller in this above-rated wind region. In OpenFAST, the rotor torque was constant at the rated torque value, whereas in the experiment, the turbine motor's proportional-integral control loop attempted to maintain that constant torque set point.

As can be seen in the rotor torque signal in Figure 10, the experiment exhibited torque variation in the low-frequency region corresponding to the wind energy range, while OpenFAST maintains a constant torque. As shown in Figure 11, the rotor speed and rotor torque were highly correlated when the torque variation was directly in phase with the rotor speed and acted to oppose rotor acceleration, essentially serving as additional rotor speed regulation in the experiment. The larger blade pitch actuation in the OpenFAST model is therefore expected since ROSCO regulates rotor speed in the above-rated region through collective blade pitch, and the additional rotor speed regulation in the experiment was not captured in the OpenFAST model. This larger blade pitch actuation leads to increased platform pitch and tower base bending around the pitch natural frequency and contributes to the discrepancy noted here. Additional factors, such as repeatability in turbulence of the wind around these frequencies and uncertainties in the aerodynamic and hydrodynamic modeling, could also play a role. The additional rotor speed variation in the OpenFAST model is likely due to the inability of ROSCO to perform the same rotor speed regulation as the experiment through blade pitch action alone. It was identified that ROSCO was tuned using a lighter than as-built rotor inertia, so the gains of the blade pitch controller are likely not as aggressive as they could be.

Further examining the blade pitch difference, we consider the relationship between rotor torque and blade pitch in this operating condition and seek to estimate how much additional blade pitch actuation would be expected in OpenFAST to apply the rotor torque variation from the experiment. The OpenFAST model shows an additional $1.3°$ of pitch actuation range, which can be taken as a rough indication of the additional blade pitch actuation required. The experiment showed a rotor torque variation range of $5.9 \times 10^6$ N·m, which represents the rotor torque variation that does not exist in OpenFAST. The estimated gradient of the Cp surface at this operating condition with respect to blade pitch ($\partial C_p / \partial \theta$) is $-0.013$, which means that effecting $5.9 \times 10^6$ N·m would require a change in blade pitch of $0.9°$. Therefore, a significant amount of the additional blade pitch actuation in the OpenFAST model may be due to not representing the generator torque variation, and a more representative rotor torque model in ROSCO could better match the experimental results.

The additional blade pitch actuation in the OpenFAST model also has an effect on the platform dynamics, especially in platform pitch, as blade pitch variation will impart thrust force variation at the nacelle. Assuming the $0.9°$ blade pitch variation identified previously, using the gradient of the Ct surface with respect to blade pitch yields a variation in thrust force of 46% of the standard deviation of rotor thrust for this condition. Since this forcing is near the platform pitch resonant frequency, the effect on platform pitch is amplified. Therefore, it is not surprising that the OpenFAST model exhibits additional platform pitch motion, as well as tower base bending motion, around the platform pitch resonant frequency. While not further quantified here, future work on the FOCAL project is considering including rotor generator models in ROSCO that can represent the as-built motor behavior and invite further investigation.

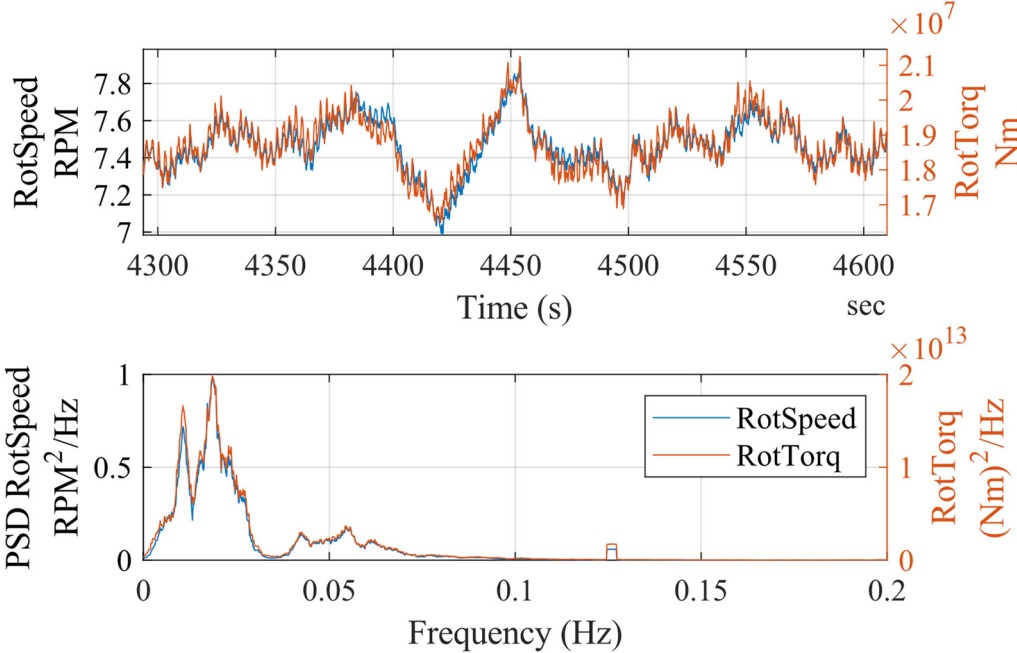

**Figure 11.** Rotor speed and rotor torque time history and PSD from experiment.

### 4.3. Performance of Floating Feedback Control Loop

The effect of the floating feedback controller was demonstrated in both the experiment and the OpenFAST model. Significant reductions in platform pitch motion and tower base bending loads were obtained around the platform pitch natural frequency, with more reduction seen in the OpenFAST model. This is partly due to the difference in blade pitch actuation caused by the constant generator torque, as described earlier, where adjustments to the blade pitch control would intuitively have a larger effect in OpenFAST, where the blade pitch controller is the only active control parameter in the above-rated region. In the experiment, the rotor speed stabilization effect due to the torque variation was still present during the floating feedback test and was similarly unaccounted for in OpenFAST. Additionally, the reduced blade pitch actuation in the experiment resulted in less platform pitch motion, which generated fewer feedback signals for the floating feedback control loop and reduced their effect.

It was also identified that retuning the filter settings to account for the shift in platform pitch angle due to the effects of the operating turbine may improve the response by more successfully filtering out the unwanted responses in higher-frequency ranges. The platform pitch frequency can shift 10–15% lower due to the effects of wind [42,43], which would affect the low-pass portion of the filter. In ROSCO, the combination of a first-order high-pass and a second-order low-pass filter is designed to isolate the response around the platform pitch natural frequency. The resulting filter is a compromise between removing unwanted feedback signals from the higher wave energy range while minimizing phase lag. Figure 12 shows the magnitude and phase of the combined filter. As shown, around the platform pitch natural frequency, the phase shift is −60 to −80 deg. As the frequency increases, the phase shift becomes more significant, causing the feedback loop to start increasing platform pitch motion at higher frequencies. The floating feedback loop is most effective around the platform pitch natural frequency, where the blade pitch correction is properly phased with the relative velocity change due to the platform pitch motion. As such, further analysis of the sensitivity of filter settings to performance of the floating feedback work is being considered.

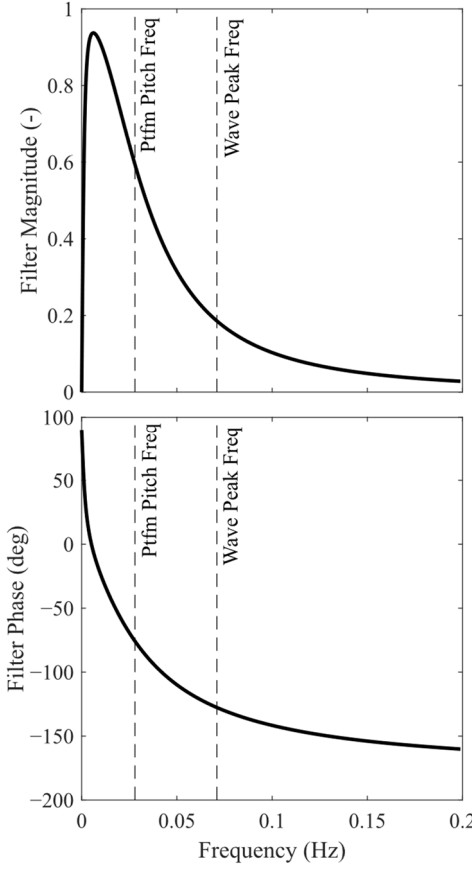

**Figure 12.** Floating feedback filter magnitude and phase.

## 5. Conclusions

The FOCAL Campaign 4 floating wind turbine experiment was simulated in Open-FAST, and results from the above-rated wind condition with the TMDs locked out were considered. Wind-only, wave-only, and combined wind/wave cases were simulated, and the effects of the ROSCO controller with and without floating feedback were investigated. Comparison of key performance metrics from the wave-only condition showed good agreement in mean values for platform surge and pitch as well as tower base bending moment and mooring tension. The OpenFAST model underpredicted the resonant surge response at the surge natural frequency due to lack of viscous excitation. Results from the wind-only case showed good agreement in platform surge and mooring tension, while the OpenFAST model predicted more blade pitch actuation than the experiment. This was partially attributed to the additional rotor speed attenuation in the experiment due to generator torque variation. ROSCO utilized a constant torque set point in the above-rated condition, which provided the torque set point for the scale model turbine. It was suggested that the rotor torque variation in the experiment, which was due to drivetrain dynamics, contributed to rotor speed regulation that was not represented in OpenFAST, resulting in larger blade pitch dynamics, rotor speed variation, and additional platform pitch and tower base moment in the simulation. In the wind/wave case, OpenFAST further underpredicted the surge resonant response, likely due to differences in aerodynamic damping and viscous effects. The constant generator torque effects from the wind-only case also applied to the wind/wave case, and the additional blade pitch dynamics resulted in similar trends, where the OpenFAST model showed larger rotor speed variation, platform pitch motion, and tower base bending moment. The effects of the floating feedback control loop were evident in both the experiment and OpenFAST, where the energy in the platform pitch motion near resonance was reduced by 32%, and the tower base bending moments reduced by 38% in the experiment, and by 59% and 61%, respectively, for OpenFAST. Both the experiment and

OpenFAST showed significant reduction in blade pitch activity near the platform resonant frequency while increasing at higher frequencies. This additional blade pitch activity in the wave energy region, in combination with the filter phase lag, started to detrimentally affect the performance of the system, and larger platform pitch motion and tower base bending moments were seen in both the experiment and OpenFAST.

**Author Contributions:** Conceptualization, M.F., A.G., R.K., D.Z., A.W. and A.R.; Data curation, E.L.; Formal analysis, M.F., E.L., R.B. and L.W.; Funding acquisition, A.V. and A.R.; Investigation, M.F.; Methodology, M.F., R.B. and L.W.; Project administration, A.R.; Software, R.B. and L.W.; Supervision, A.V., A.G. and R.K.; Visualization, E.L.; Writing—original draft, M.F.; Writing—review and editing, A.G., R.K., R.B., L.W., D.Z., A.W. and A.R. All authors have read and agreed to the published version of the manuscript.

**Funding:** This work was authored in part by the University of Maine for the U.S. Department of Energy (DOE) under Award No. DE-AR0001183. Funding provided by the U.S. Department of Energy Advanced Research Projects Agency-Energy (ARPA-E) under the Aero-dynamic Turbines, Lighter and Afloat, with Nautical Technologies and Integrated Servo-control (ATLANTIS) program. This work was authored in part by the National Renewable Energy Laboratory, operated by Alliance for Sustainable Energy, LLC, for the U.S. Department of Energy (DOE) under Contract No. DE-AC36-08GO28308. The views expressed in the article do not necessarily represent the views of the DOE or the U.S. Government. The U.S. Government retains and the publisher, by accepting the article for publication, acknowledges that the U.S. Government retains a nonexclusive, paid-up, irrevocable, worldwide license to publish or reproduce the published form of this work, or allow others to do so, for U.S. Government purposes.

**Institutional Review Board Statement:** Not applicable.

**Informed Consent Statement:** Not applicable.

**Data Availability Statement:** The experimental data referenced in this study, inclusive of all four experimental campaigns and the OpenFAST model, are available on the Department of Energy Atmosphere to Electrons website at https://a2e.energy.gov/projects/focal.

**Acknowledgments:** The authors would like to acknowledge support from the staff and students at the University of Maine's Wind/Wave Ocean Engineering Laboratory for their efforts in conducting the FOCAL experimental work.

**Conflicts of Interest:** The authors declare that they have no known competing financial interest or personal relationships that could have appeared to influence the work reported in this paper.

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
