# Peer review of "Wind/Wave Testing of a 1:70-Scale Performance-Matched Model of the IEA Wind 15 MW Reference Wind Turbine with Real-Time ROSCO Control and Floating Feedback"

_machines, doi:10.3390/machines11090865_

Round 1

Reviewer 1 Report

This is a well-written paper containing meaningful results. However, a number of points need clarification and certain statements require further justification. Following comments for authors' consideration.

(1)  Table 7 is split over 2 pages. Similarly for Table 8 and Table 10. In Table 7, ‘Frequency Range [Hz[’ should be modified to ‘Frequency Range [Hz]’.

(2)  There are two Table 6.

(3)  Please provide a detailed analysis of the discrepancies between the OpenFAST results and the experimental results shown in Figures 9 and 10, particularly focusing on the frequency range prior to 0.05 Hz.

(4)  The format of references needs attention. There are several errors.

Good enough except for several editing mistakes.

Author Response

Thank you for your comments. I have included "comments" in the revised word document as well as the attached response.

Reviewer 2 Report

This study and presentation of results is scientifically sound, with good agreement found between simulated and experimental results for across many but not all parameters. Where disagreement was found, the authors suggested plausible reasons for differences between code and tank tests. The authors should be commended for this contribution to the wind energy research community, especially for their efforts and success in providing data that is open-source. 

My only suggestions for improvement are therefore minor and related to small details in the text. My list of recommendations is provided here:

- Line 80: affect vs effect. 

- Figures 1 and 2 are rather grainy. Check image quality throughout.

- Figures 4, 5, and 11: This may be a personal preference, but I find your y-axis labeling to be confusing. If a PSD is included as a subplot, then the axis should label it as such. More specifically, I suggest in Figure 4 that the bottom subplot's y-axis be labelled with "PSD Wind Speed, U" to differentiate it from the top subplot which is actually just the wind speed. (Figures 7, 8, 9 do not have this confusion, as all their subplots are PSDs.)

- Line 253: "cable umbilical" reads awkwardly. I recommend "umbilical cable" instead. Change should occur at multiple locations in the paper. 

- Table 7: The second bracket around [Hz] is incorrect. 

Author Response

Thank you for your comments. I have included "comments" in the revised manuscript as well as the attached response document.
